# Healthy Life Habits in Caregivers of Children in Vulnerable Populations: A Cluster Analysis

**DOI:** 10.3390/ijerph21050537

**Published:** 2024-04-25

**Authors:** Moisés Mebarak, Juan Mendoza, Duban Romero, José Amar

**Affiliations:** 1Human Development Research Center, Universidad del Norte, Barranquilla 080020, Colombia; mmebarak@uninorte.edu.co (M.M.); jamar@uninorte.edu.co (J.A.); 2Department of Psychology, Universidad del Norte, Barranquilla 080020, Colombia; cjarango@uninorte.edu.co

**Keywords:** healthy lifestyle habits, attitudes, childhood, cluster analysis, Latin America

## Abstract

Intervention programs aimed at mitigating the effects of chronic noncommunicable disease (CNDs) focus on promoting healthy lifestyle habits (HLH), especially in the early stages of life. Because of this, different typologies of caregivers have been identified according to HLH during middle childhood and adolescence. However, the available studies have focused on aspects such as nutrition, physical activity, and rest, ignoring other HLHs that are equally important for children’s well-being. Likewise, few studies address HLH during the first five years of life and how caregivers affect children’s health. In a sample of 544 caregivers of children aged zero to five years from low-income Colombian communities, we established a typology of attitudes toward different HLHs. The results indicate the presence of three clusters that grouped caregivers with (1) positive attitudes toward all HLHs, (2) toward some HLHs, and (3) relatively low positive attitudes toward all HLHs. Membership in clusters with less positive attitudes toward HLHs was also found to be associated with low educational levels and living in rural areas. This study detected profiles of caregivers who may have unhealthy lifestyles, so the results would allow social workers to design differential interventions on HLHs in non-industrialized countries.

## 1. Introduction

Chronic non-communicable diseases (CND) such as obesity, different types of cancer, or diabetes are the leading cause of death in the world [1,2]. One of the CNDs that has shown an exponential increase in the last century is obesity [3,4,5]. According to the World Health Organization [6], in 2016, the global prevalence of obesity in adults was 13%, representing an almost threefold increase in prevalence rate since 1975, with the Americas being the region of the world with the greatest increase in obesity rates since the 2000s [6]. By 2016, 340 million children and adolescents aged 5–19 years and 40 million children under 5 years were overweight or obese [2,6]. Weight gain during these life stages is associated with an increased likelihood of obesity, type 2 diabetes, and premature death in adulthood [6].

In view of the negative effects of CNDs on people’s health and well-being, the WHO has established multiple prevention plans that aim to reduce the impact of these conditions [7,8,9]. Most programs aimed at mitigating the effects of CNDs seek to promote HLHs [10]. This is because unhealthy lifestyle habits such as poor-quality diet, low physical activity, excessive alcohol or tobacco consumption, or poor sleep quality, can increase the risk of developing CNDs [2,5,11,12,13]. In contrast, more healthy behaviors are associated with better physical, psychological functioning and higher quality of life [14,15,16,17,18,19,20,21,22,23,24,25]. Evidence shows that low fruit and vegetable consumption and high consumption of fatty foods, sugary snacks, sweets, sedentary behaviors (screen and TV time > 2 h/day), and low sleep time (<10 h/day) are positively associated with overweight and/or obesity [5,26]. Healthy lifestyle habits (HLHs) usually develop mainly during childhood and adolescence and are maintained into adulthood [12,27,28,29,30]. Therefore, prevention and intervention during the early stages of life is essential to establish the lifestyle habits that will be present during the rest of life [3,19,24,27].

Especially in their first years of life, children need to be constantly supervised by others to ensure their survival and well-being [31,32]. During this stage of life, children’s behaviors are strongly influenced by the attitudes and living habits of their caregivers, who may facilitate (or limit) certain activities [3,4,5,24,30], model new patterns of behaviors [5,25,28,30,33,34], or instill life values [4,30,35]. Given that caregivers are a key factor in the acquisition of children’s life habits, many intervention programs that seek to address the health and well-being of young children have focused on changing caregivers’ parenting practices, either through psychoeducation or by making them an integral part of the change process [5,28,29,30,31,36,37].

People from disadvantaged social backgrounds have a shorter life expectancy and suffer from disabilities for most of their lives [2,23,33,35,38,39,40]. Related to this, the previous research has found evidence that the association between socioeconomic status (SES) and health is partially mediated by living habits [13,40]. Even though populations of low SES are more likely to acquire and maintain unhealthy lifestyle habits, a large proportion of intervention programs focus on populations in higher income countries [2,3,23,36]. Moreover, interventions that aim to promote the acquisition of healthy lifestyles do not necessarily show the same effectiveness in populations of low and high socioeconomic status [23,38].

Although many of the programs available to improve the diet, sleep patterns, or physical activity of children and adolescents recognize the importance of providing participants with relevant knowledge about HLH [4,5,24,31,35,36,37], very few attempt to change the home environment or directly involve caregivers to facilitate children’s lifestyle change [4,31,34,36]. This is important because, traditionally, intervention programs target the family or school environment [28,38], ignoring other contexts that are also part of the children’s life space [25,30,32,33] or providing information to caregivers about healthy eating, how much sleep children should get, or how much exercise they should do per day. However, the research has shown that these approaches have a limited effect on the children’s and caregivers’ behaviors [29,34,38,39]. In view of this, more recent interventions have opted for a generalized change in the spaces where children develop and the direct involvement the family environment, mainly through caregiver’s health-promoting behaviors [2,5,28,32,36,39]. That said, it is necessary to recognize that in the contexts of economic and social hardship, the change in the children’s life space or the direct involvement of caregivers is significantly constrained by macro-social factors that can hardly be addressed in the context of psychosocial or behavioral interventions [2,4,25,36,40]. Thus, the implementation of effective and massive strategies that promote HLH in these vulnerable communities requires a detailed profiling that considers the specific characteristics, needs, and resources of the people living in these conditions [5,23,31,33].

The previous studies have established different typologies of HLHs based on diet, physical activity, sleep quality or duration, or time spent by children in front of screens [19,41]. Although established typologies may vary from study to study, broadly speaking, HLHs can be classified into healthy, unhealthy, and mixed patterns [41]. A healthy pattern refers to good diet quality, high physical activity, few sedentary behaviors, and adequate sleep. In contrast, an unhealthy pattern would correspond to a low-quality diet, low physical activity, many sedentary behaviors, and inadequate sleep. Mixed patterns included concurrent healthy and unhealthy behaviors [21,41,42,43,44]. Other studies have found that HLHs can be grouped into more than four classes [45,46]. HLH profiles have also been identified from caregiver parenting patterns, although these studies have been limited to feeding practices [47,48].

In the Latin American context, typologies of HLHs have already been carried out [49,50,51]. Infante-Grandón et al. [52] identified three profiles in adolescents from southern Chile. They presented a healthy group characterized by a high frequency of physical activity, little time in front of the screen, and high consumption of vegetables and legumes. The second group was characterized by low physical activity, low consumption of vegetables and legumes, and high exposure to the screen. The third group had low physical activity, low screen time, and low consumption of vegetables and legumes [52]. Likewise, Varela Arévalo et al. [53] identified two HLH profiles in Colombian schoolchildren aged eight to twelve years based on their dietary habits, digital entertainment consumption, and physical activity. A first cluster was characterized by low physical activity, while the second cluster grouped children with high digital entertainment consumption [53].

Multiple sociodemographic aspects, such as income level, gender and age of the child, and area of residence can have a significant effect on healthy living habits. Children from disadvantaged schools and lower-income families tend to have unhealthy eating habits compared to children from high-income schools and families [5,36,38,52,53,54,55]. Likewise, children who grow up with one or neither parent tend to report fewer positive attitudes toward healthy habits, as well as behaviors such as smoking, drug use, violence, low physical exercise, and diet of poor nutritional quality compared to those who grow up under the care of both parents [56]. In terms of gender, male children tend to present with a higher number of unhealthy lifestyle habits compared to females, especially in terms of diet quality [14,45,46,57]. Older children tend to be more physically active [58] and have more regular eating patterns [57], although age is also related to a higher frequency of sedentary activities, such as video games and TV [59]. Furthermore, children living in urban communities are more likely to be overweight compared to those living in rural communities [52,60,61].

Caregiver education has also been associated with children’s healthy lifestyle habits, especially when it comes to eating habits [11,25,42,45,57,61,62]. For example, Sanmarchi et al. [58] found that adherence to a Mediterranean diet is positively associated with the fathers’ educational level, while children’s screen time is related to a lower mothers’ educational level. Similarly, children of highly educated parents consume milk, fruits, and vegetables more frequently, engage in more physical activity, and spend fewer hours in front of the TV screen [55,63]. This is associated with more educated caregivers being more likely to correctly perceive diet quality [64] and to favor habits, such as not having television in the children’s room, that avoid excessive hours in front of screens [59].

As a contribution to the ongoing efforts to study lifestyle habits in the early stages of life, the present research examined the attitudes towards HLH in a sample of caregivers for children aged zero to five years from low socioeconomic communities in Colombia, in South America. There are several reasons why this is important. First, studies that have established typologies of HLH have mainly considered aspects such as nutrition, physical activity, and resting patterns [18,19,21,41,42,43,44,45,46], overlooking other lifestyle habits that are key to the health and wellbeing of children and adolescents (e.g., sexuality or substance use). On the other hand, except for a few studies [49,50,51,52,53], most of the research into HLH has little participation from individuals from the Global South and much less from people who have low economic income [2,36], thus, limiting the knowledge available about low SES communities that are more likely to have unhealthy lifestyles and reducing the effectiveness of intervention programs that aim to improve the wellbeing and health of people living in these conditions [2,38]. The examination of the attitudes towards HLH in low SES communities would, therefore, open possibilities for carrying out intervention programs with more reliable designs and higher effectiveness rates.

### The Present Study

A hefty number of studies related to HLH focus on middle childhood and adolescence [18,21,43,44,45,49,50,51]. However, there are no known studies that establish the profiles of child caregivers around their attitudes towards HLH in Latin America. This is particularly relevant in vulnerable communities where children tend to present a higher number of unhealthy lifestyle habits [52,53,54,55]. In the present study, we sought to establish a typology of caregivers of children aged zero to five years considering their attitudes towards HLHs. To carry this out, we recruited caregivers from vulnerable communities of the Colombian Caribbean region and evaluated their attitudes toward several HLH, such as physical activity, nutrition, and sexuality. We then examined whether the data clustered naturally using the k-means algorithm and then compared group means using multiple one-way ANOVA models. Given that the previous studies have reported that children’s HLH can be classified as healthy, unhealthy, and mixed patterns, we expected to find a three-cluster solution that pointed towards positive, not so positive, and mixed caregivers’ attitudes towards HLHs. 

Additionally, we examined whether the membership of a given group is associated with the participants’ demographic characteristics, such as occupation, kinship, or the sex and age of the children. As there is evidence that the educational level, living area, and family type might influence children’s HLH [11,25,42,45,55,56,57,58,59,61,62,63,64], we expected caregivers with the most positive attitudes towards HLHs to live in rural communities, have a high level of education, and not be part of single-parent families. However, we did not expect that the sex or age of the children were related to the caregiver’s membership in a given cluster [14,45,46,57,58,59].

## 2. Methods

### 2.1. Procedure

The caregivers were recruited through district agencies and the Colombian Caribbean childcare units. The childcare units summoned all the caregivers of the affiliated children to familiarize them with the research project. Among the project participants were those caregivers who had expressed complete willingness to participate in the research. Before starting data collection, the participants read and signed an informed consent form that included all information related to the present study. The data collection was carried out during August and September 2022 in the childcare centers (CDI) of the municipalities of Repelón, Luruaco, Manatí, Santa Lucía, and Barranquilla, belonging to the Colombian Caribbean Coast. The surveys were administered by three research assistants of the project. The survey included other measures that were not relevant for the present study.

### 2.2. Instruments

*Demographic Questionnaire.* Participants were asked about their age in years of caregivers and the children in their care, biological sex, family type, relationship to the child, educational level, main occupation (student, worker, homemaker, etc.), and their income level.

*Healthy Living Habits Scale (HLHS).* The HLHS is a 47-item instrument developed by the Pan American Health Organization (PAHO) to assess healthy lifestyles. In this study the HLH was adapted to measure caregivers’ attitudes toward relationships with others (7 items; α = 0.91), physical activity (3 items; α = 0.86), rest (4 items; α = 0.89), nutrition (4 items; α = 0.71), oral care (4 items; α = 0.81), sexuality (5 items; α = 0.86), substance use (5 items; α = 0.94), optimism (5 items; α = 0.86), environmental care (4 items; α = 0.90), and mobility (6 items; α = 0.92). The mobility scale items were adapted so that they could be applied in rural samples. HLHS responses are given on a 5-point scale (1 = strongly disagree; 5 = strongly agree). This instrument has been implemented in past studies with Colombia samples [65]. Psychometric analyses of the HLHS were conducted to ensure its validity and reliability (the analyses are available at https://bit.ly/41j9E4Z, accessed on 15 January 2024). Higher scores indicate more favorable attitudes toward childcare.

### 2.3. Data Analysis

First, missing values in the HLHS scales were imputed using the predictive mean matching method, implemented in ‘mice’ [66] in R [67], since its use is recommended for multivariate imputation of continuous data. Descriptive measures and Pearson’s correlation coefficients were then calculated for the HLHs. Participant profiling was performed using cluster analysis. The Euclidean distance matrix was used, and the K-means clustering method was performed. Considering that K-means requires setting several factors a priori, the ‘NbClust’ package [68] was implemented to determine the most appropriate number of profiles to extract. After the profiles were established, group comparisons were conducted using multiple one-way ANOVAs to establish the significance of differences between groups, and the χ2 test of independence was used to examine whether membership in any clusters is related to the participants’ demographic characteristics.

## 3. Results

A total of 544 caregivers of children aged zero to five years participated in this study, 513 females (94.3%), 25 males (4.6%), and 6 who did not indicate their sex (1.1%). The ages of the caregivers ranged from 18 to 62 years (*M* = 29.28, SD = 8.53). Of the total sample, 53.7% of the caregivers were from extended families, 37% from nuclear families, and the remaining 9.3% from single-parent families. Of the caregivers, 86.9% were biological mothers of the children in their care, 6.36% were their grandmothers, and 6.47% were biological fathers, uncles, aunts, among other types of kinship. Regarding educational level, 96.16% did not have a university education, while the remaining 3.84% had a higher education degree. Finally, 76.62% of the participants were engaged in household chores, 17.63% were engaged in work, and 5.75% were students.

Descriptive analyses showed that caregivers hold favorable attitudes toward most healthy living habits, especially optimism (*M* = 4.49, SD = 0.54), mobility (*M* = 4.46, SD = 0.59), and environmental care (*M* = 4.44, SD = 0.59). Examining the correlations between scales of the HLH shows that most cases have moderate or strong positive correlations (see Table 1).

When examining the distances between subjects, four individuals were identified as being significantly different from the others. Therefore, it was decided to eliminate them from the cluster analysis. The initial analysis of the Euclidean distance matrix indicated that the data were grouped into three clusters, which explained 73.53% of the original information (see Figure 1). To assure that the three-clusters solution was the optimal configuration for our data, we vary the number of clusters from one to five and evaluate their fit with multiple statistical indices. Because most of the indices agreed on the existence of three clusters in the data, we retained three profiles of caregivers according to their attitudes to the different HLHs.

Cluster 1 (n = 213) agglomerated those participants who have more positive attitudes towards the different HLHs compared to the other groups. Cluster 2 (n = 78) showed greater variability in attitudes toward each HLH, with those belonging to this group having positive attitudes toward interpersonal relationships, rest, healthy sexual relationships, and optimism and showing fewer positive attitudes toward substance use avoidance and physical activity. Finally, cluster 3 (n = 307) identified as caregivers who, although they showed positive attitudes towards the different HLH, exhibited attitudes that were less positive than in the other clusters (see Figure 2). We further examined the mean differences in the attitudes towards each HLH among profiles using multiple one-way ANOVA models. The results indicated that the profiles differ significantly from one another in terms of between HLHs, most of these mean differences having a large size effect, F (2, 590) > 139.210, *p* < 0.001, η_2_ > 0.320.

After the three caregiver profiles were established, we proceeded to examine whether each participant’s membership in any of the profiles was associated with his or her demographic characteristics. The results showed that the caregiver’s educational level was associated with membership in the different clusters, χ^2^(2) = 7.11, *p* = 0.029, and V = 0.10. Caregivers with no college education tended to belong to the third cluster, whereas those with a college degree tended to be classified in the first cluster. Furthermore, the results indicated that area of residence was associated with membership in the different clusters, χ^2^(2) = 10.71, *p* = 0.005, and V = 0.13. The caregivers living in rural areas were clustered to a greater extent in cluster 3, while people coming from urban areas tended to be in cluster 1 (see Figure 3). Membership in the different clusters was not associated with occupation, kinship, family type, sex, or age group of the caregiver, *p* ≥ 0.05.

## 4. Discussion

In the present study, we aimed to establish a typology of caregivers from low SES Colombian communities based on their attitudes towards different HLHs. Consistent with the previous work, our results suggest the existence of three profiles of caregivers using the k-means clustering algorithm. Additionally, multiple group comparisons revealed large differences among profiles in all HLH attitudes assessed. We also found evidence of associations between cluster membership and some demographic variables; caregivers who had not completed university and who lived in rural areas tended to be in the group with the least positive attitudes towards all HLH. Overall, these results provisionally support the existence of three profiles of caregivers in low SES communities from the Colombian Caribbean.

Results of the k-means clustering algorithm indicate that participants can be grouped in three groups based on their attitudes toward HLHs: one class where caregivers present positive attitudes toward all HLH, another class where caregivers present fewer positive attitudes toward all HLH, and another mixed class where caregivers have positive attitudes toward most HLH except for physical activity and substance use. This is consistent with the previous research performed by D’Souza et al. [41] and others [21,42,43,44,52], in which eating and physical activity habits of children and adolescents were grouped into healthy, unhealthy, and mixed patterns. It should be noted that unlike most of the previous research [19,21,41,42,43,44,45,46,47,48,49,50,51,52,53], in the present study, the profiles were constructed based on caregivers’ attitudes towards HLHs and they did not consider children’s habits directly. The similarity between the profiles may be because caregivers’ attitudes are related to children’s living habits, especially during their first years of life [37,55,56,69].

Regarding demographic variables, membership in each of the clusters was weakly linked with the caregiver educational level. Participants without a college degree tended to belong to the cluster with the least positive attitudes toward the different HLH. This is in line with previous studies, in which low parental educational level is associated with unhealthy eating habits, sedentary lifestyle, obesity, or irregular meal patterns [12,42,45,57]. Likewise, the area of residence showed a small association with membership in the different clusters. The caregivers from rural areas were most likely to belong to cluster 3, with the least positive attitudes towards the different HLH. This contrasts with the findings of the previous research [52,60,61], according to which children living in urban communities are more likely to be overweight and maintain less healthy HLHs compared to those living in rural communities. Regarding the sex and age of the children, no statistically significant association was found in the clusters. This agrees with observations from some previous studies where neither the sex nor age of children were related to their HLH [16,19,47,63,70,71].

### Strengths and Limitations

The main strength of this study lies in the fact that for the first time, it assesses caregivers’ attitudes towards different HLHs beyond diet quality, physical activity, and sleep quality or duration. In addition, the study focuses on the caregivers of children under five years of age from vulnerable Latin American communities, a population that, despite being more prone to maintain unhealthy lifestyle habits, has not been studied. The main limitation lies in the fact that only the caregivers’ attitudes towards HLH were measured, and the participants’ habits or their children’s health status, such as body mass index, were not measured directly. Future research should jointly study caregivers and children to determine the effects of parenting practices on the health of young children. Moreover, further research should be conducted to expand our knowledge of these vulnerable populations, especially in the Latin American context. Furthermore, it is suggested to focus interventions on groups of caregivers with mixed attitudes toward different HLHs.

## 5. Conclusions

To date, studies that have proposed typologies of caregivers have focused on children’s patterns of feeding, physical activity, and rest during middle childhood or adolescence, while the implication of others HLHs remains scarce. In the present study, we established a typology of caregivers of children aged zero to five from vulnerable populations on the Colombian Caribbean coast. We found that caregivers can be categorized into three groups based on their attitudes towards different HLHs: (1) caregivers who have a positive attitude towards all HLHs, (2) who have positive attitudes towards some HLHs, and (3) who have a low positive attitude towards all HLHs. The results also indicated that group membership is weakly associated with caregiver educational level and place of residence. Overall, we conclude that caregivers in Colombian vulnerable populations can be clustered in three groups based on their attitudes towards the HLHs. However, more research is needed to clarify the validity of the present typology and to explore its relationship with the physical health and psychological wellbeing of children.

## Figures and Tables

**Figure 1 ijerph-21-00537-f001:**
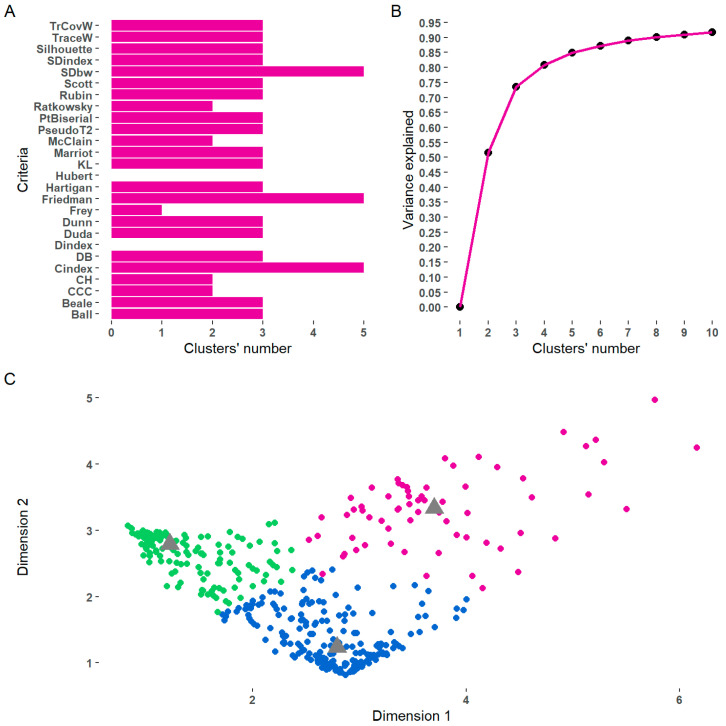
Number and cluster extraction. (**A**) shows the cluster numbers that should be extracted according to several statistical criteria [68]. (**B**) illustrates the cumulative variance explained when the cluster number is extracted. In (**C**), distribution observations are shown in a bidimensional plane where color indicates the cluster membership. In (**C**), triangles represent centroids of each cluster.

**Figure 2 ijerph-21-00537-f002:**
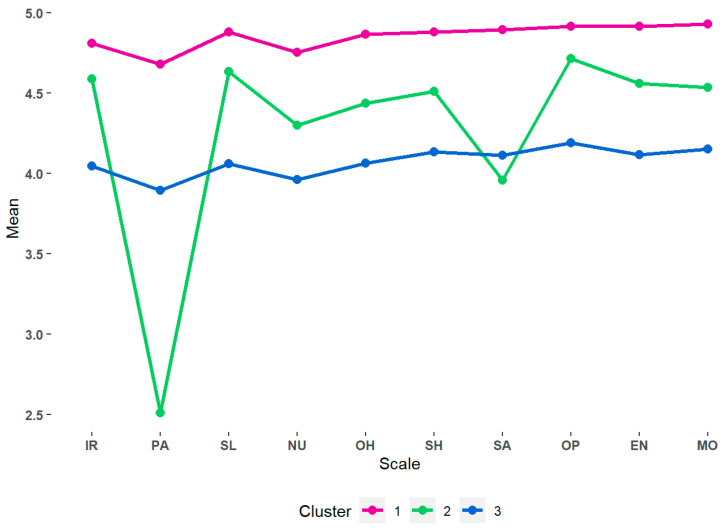
Means score for each cluster. All the statistical differences between groups for each variable were significant. IR = interpersonal relationships; PA = physical activity; SL = sleep; NU = nutrition; OH = oral health; SH = sexual health; SA = substance abuse; OP = optimism; EN = environmental care; MO = mobility.

**Figure 3 ijerph-21-00537-f003:**
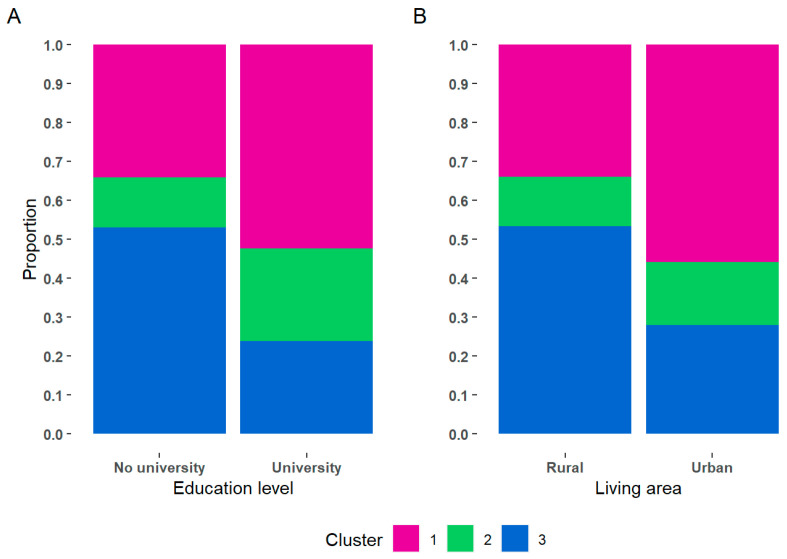
Associations between cluster membership and demographics. In each bar, the amplitude of the colored area indicates the proportion of individuals from a cluster who possess the specific demographic characteristic. For example, in (**A**), the blue area indicates the proportion of individuals from cluster 3 who either have a college education or do not while in (**B**) blue area represents the proportion of individuals from cluster 3 who either live in rural area or do not.

**Table 1 ijerph-21-00537-t001:** Descriptive statistics and Pearson’s correlation coefficients of the HLHs.

	*M* (SD)	RS	PA	RE	NU	OC	SE	SU	OP	EN	MO
RS	4.37 (0.57)	1									
PA	3.98 (0.95)	0.25	1								
RE	4.40 (0.57)	0.82	0.29	1							
NU	4.27 (0.61)	0.69	0.34	0.76	1						
OC	4.37 (0.56)	0.69	0.44	0.76	0.74	1					
SE	4.43 (0.60)	0.69	0.19	0.71	0.64	0.67	1				
SU	4.35 (0.75)	0.57	0.21	0.61	0.59	0.51	0.72	1			
OP	4.49 (0.54)	0.76	0.21	0.77	0.69	0.75	0.77	0.61	1		
EN	4.44 (0.59)	0.68	0.25	0.67	0.56	0.62	0.59	0.47	0.71	1	
MO	4.46 (0.59)	0.64	0.27	0.66	0.59	0.66	0.62	0.49	0.74	0.85	1

Note. All hypothesis tests were statistically significant (*p* < 0.001); RS = relationships with others; PA = physical activity; RE = rest; NU = nutrition; OC = oral care; SE = sexuality; SU = substance use; OP = optimism; EN = environmental care; MO = mobility; *M* = arithmetic mean; SD = standard deviation.

## Data Availability

The data collected for this study are available at https://bit.ly/41j9E4Z (accessed on 15 January 2024).

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
