# Peer review of "Healthy Life Habits in Caregivers of Children in Vulnerable Populations: A Cluster Analysis"

_ijerph, 2024, doi:10.3390/ijerph21050537_

Round 1
Reviewer 1 Report
Comments and Suggestions for Authors
The article focuses on examining attitudes towards various health behaviors of caregivers of children under 5 years of age from low-income Colombian communities. This problem is very important considering the advisability of starting preventive activities at the early stages of life aimed at establishing correct lifestyle habits that will be present for the rest of life. In such activities, the role of children's caregivers is most important, hence learning about their attitudes towards healthy lifestyle habits (HLH) will indicate to whom appropriate intervention activities should be directed. The study participants were 544 caregivers of children aged zero to five years.
The study used a demographic questionnaire containing questions about age in years of caregivers and children in their care, biological sex, family type, relationship to the child, educational level, main occupation, and income level. The basic tool was the Healthy Living Habits Scale (HLHS) - a 47-item instrument to assess healthy lifestyles. Using cluster analysis, three groups of caregivers were distinguished with (1) a positive attitude towards all HLHs, (2) towards some HLHs and (3) a relatively low positive attitude towards all HLHs. Less positive attitudes were subsequently shown to be associated with low levels of education and living in rural areas. Thus, the goal with which the study was initiated was achieved.
I have two minor comments on the article:
There is no description of how the study participants were recruited (only information about who conducted the study), so it is impossible to assess how representative the group is.
The titles of tables and figures are too laconic, they should describe exactly what is presented in them. All abbreviations should be explained. Meanwhile, in Table 1 there is no explanation of the abbreviations AC and DE. Furthermore, if Table 1 contains correlations of all HLHs, why was the PA=Physical activity variable only correlated with RE=Rest.
Author Response
Reviewer 1
The article focuses on examining attitudes towards various health behaviors of caregivers of children under 5 years of age from low-income Colombian communities. This problem is very important considering the advisability of starting preventive activities at the early stages of life aimed at establishing correct lifestyle habits that will be present for the rest of life. In such activities, the role of children's caregivers is most important, hence learning about their attitudes towards healthy lifestyle habits (HLH) will indicate to whom appropriate intervention activities should be directed. The study participants were 544 caregivers of children aged zero to five years.
The study used a demographic questionnaire containing questions about age in years of caregivers and children in their care, biological sex, family type, relationship to the child, educational level, main occupation, and income level. The basic tool was the Healthy Living Habits Scale (HLHS) - a 47-item instrument to assess healthy lifestyles. Using cluster analysis, three groups of caregivers were distinguished with (1) a positive attitude towards all HLHs, (2) towards some HLHs and (3) a relatively low positive attitude towards all HLHs. Less positive attitudes were subsequently shown to be associated with low levels of education and living in rural areas. Thus, the goal with which the study was initiated was achieved.
I have two minor comments on the article:
There is no description of how the study participants were recruited (only information about who conducted the study), so it is impossible to assess how representative the group is.
Authors: We greatly appreciate your thorough review of the manuscript. We have proceeded to add information on how participants were recruited in the procedure section.
The titles of tables and figures are too laconic, they should describe exactly what is presented in them. All abbreviations should be explained. Meanwhile, in Table 1 there is no explanation of the abbreviations AC and DE. Furthermore, if Table 1 contains correlations of all HLHs, why was the PA=Physical activity variable only correlated with RE=Rest.
Authors: Thank you for your recommendation. There was a typing error in Table 1, so the mentioned abbreviations were adjusted. In Figure 1, a brief description was added to facilitate the reader's understanding of the scatter plot. In Figure 2, the title was adjusted to provide more informative about the presented information. In Figure 3, a note was added explaining what each colored area represents.
Reviewer 2 Report
Comments and Suggestions for Authors
Thank you for the opportunity to review this work. The authors have aimed to establish a typology of caregivers of children five years or below considering their attitudes towards healthy lifestyle habits (HLH). It is interesting that the authors looked at the attitudes of caregivers of children towards HLH. It would have been interesting to examine this in children as well which the authors mention as one of the study limitations. Some updates to the manuscript will help to improve clarity.
· Methods: In the first section under “participants”, a lot of detail is presented which would fit better in the results section. Here, the authors are describing the population demographics and should consider moving the details to the first paragraph of the results and condense this section. Most of this shows descriptive statistics.
· The information presented under “procedure” should be the first section of the methods because it describes how the sample was recruited.
· Please further describe how participants were recruited.
· Line 116: most of the participants are females. Would be helpful to further explain this? Was participant recruitment targeted towards getting an equal representation of males and females?
· Line 130: is the HLHS scale most commonly used in the other studies?
· Line 176-177: did the authors consider including these 4 participants in the primary analysis and then excluding them in a sensitivity analysis?
Comments on the Quality of English Language
Minor editing to English language needed.
Author Response
Reviewer 2
Thank you for the opportunity to review this work. The authors have aimed to establish a typology of caregivers of children five years or below considering their attitudes towards healthy lifestyle habits (HLH). It is interesting that the authors looked at the attitudes of caregivers of children towards HLH. It would have been interesting to examine this in children as well which the authors mention as one of the study limitations. Some updates to the manuscript will help to improve clarity.
Authors: We appreciate your thorough review of the manuscript as well as the recommendations provided to enhance it.
Methods: In the first section under “participants”, a lot of detail is presented which would fit better in the results section. Here, the authors are describing the population demographics and should consider moving the details to the first paragraph of the results and condense this section. Most of this shows descriptive statistics.
Authors: Thank you for your recommendation. Done.
The information presented under “procedure” should be the first section of the methods because it describes how the sample was recruited.
Authors: Thank you for your recommendation. Done.
Please further describe how participants were recruited.
Authors: Thank you for your request, which aligns with what was requested by the other reviewer. A brief description of the recruitment strategy was added to the procedure description.
Line 116: most of the participants are females. Would be helpful to further explain this? Was participant recruitment targeted towards getting an equal representation of males and females?
Authors: In this region of Colombia, the majority of caregivers are women due to the existence of deeply ingrained traditional gender roles. Typically, women in this region are the ones who stay at home taking care of children. We did not implement any strategy to balance the participation of men because this study is part of a broader project aimed at strengthening childcare practices. Therefore, the most important factor was that the participant was primarily responsible for the child's care. With the results of this study, it is hoped that future interventions will include a gender equity component in promoting childcare practices.
Line 130: is the HLHS scale most commonly used in the other studies?
Authors: The instrument used to measure healthy habits in this study is not widely known internationally; however, there are previous studies in Colombia where it has been used. In the instrument description, we provided a reference to one of these studies.
Line 176-177: did the authors consider including these 4 participants in the primary analysis and then excluding them in a sensitivity analysis?
Authors: "Those four participants were not included in the sensitivity analysis; rather, they were excluded from the outset because they were affecting the variability within each cluster and consequently the centroids. This affects the final cluster estimation. In this regard, it was not necessary to include these observations in the sensitivity analysis since from the creation of the cluster, these observations were impacting the analyses."
Round 2
Reviewer 2 Report
Comments and Suggestions for Authors
All previous concerns have been addressed.
Comments on the Quality of English LanguageMinor editing to English language needed.